# Experimental and Numerical Analysis of the Concrete Maturation Process with Additive of Phase Change Materials

**DOI:** 10.3390/ma15134687

**Published:** 2022-07-04

**Authors:** Mahmoud Hsino, Tomasz Jankowiak, Józef Jasiczak

**Affiliations:** 1Department of Civil Engineering, Academy of Applied Sciences Stanisław Staszic in Piła, Podchorazych 10, 64-920 Pila, Poland; mhsino@puss.pila.pl; 2Institute of Structural Analysis, Poznan University of Technology, Piotrowo 5, 60-965 Poznan, Poland; 3Institute of Civil Engineering, Poznan University of Technology, Piotrowo 5, 60-965 Poznan, Poland; jozef.jasiczak@put.poznan.pl

**Keywords:** phase change materials, laboratory tests, numerical simulations, maturation of concrete

## Abstract

The article presents selected types of phase change materials (PCM) and their properties in terms of applications in various fields of science such as construction and concrete technology. The aim of the article is to present a comparative analysis between the results of the laboratory tests and numerical simulations. The analysis contains two types of PCM (powder and in liquid), which were dosed in a hybrid system to the concrete mix. The purpose of using PCM is to allow the technological barrier to be exceeded in hot and dry climate conditions, enabling the construction of non-cracking concrete structures. The paper presents a parametric analysis of the influence of various modeling elements on the obtained results. The procedure of generating and absorbing heat caused by the applied PCM was also implemented using user subroutine into finite element code (Abaqus/Standard). The numerically obtained results are consistent with the experimental results. The presented results demonstrate that the use of PCM improves the conditions of concrete maturation by reducing the average temperature of the mixture in its entire volume.

## 1. Introduction

The process of concreting in dry and hot conditions differs significantly from the performance of these works in a temperate climate. At a concrete temperature above 32 °C, blowing wind and insufficient air humidity intensify the phenomena of accelerated evaporation of water from the concrete mix. The process is exponential, and even small changes in temperature cause a significant loss of water from the maturing young concrete, which reduces its compressive strength and causes the appearance of shrinkage cracks [1,2,3,4]. Then, significant temperature and temperature gradients form inside the concrete, caused on the one hand by the influence of atmospheric factors and, on the other hand, by the cement’s hydration. Therefore, it is necessary to maintain the external and internal care of the concrete mix immediately after its arrangement in the structure, to ensure the continuity of the hydration process and concrete setting, and to avoid scratches and cracks in the concrete element caused by loss of mixing water and increment of the temperature inside the concrete element.

A necessary condition for the continuation of the correct mixture maturation process is to maintain the relative humidity inside the concrete higher than 80%. The essence of maintenance is to keep the concrete saturated with water or as saturated as possible, until the spaces originally filled with water in fresh cement slurry are filled to the desired degree by cement hydration products [2].

If the concrete element is constructed without external barriers, it can freely expand during heating and contract during cooling, without causing stresses [3]. In practice, however, concrete is almost always constrained to some extent by adjacent structures (external constraint) or internally due to the temperature increment in the structural member itself (internal constraint). Massive structures, such as dams, tend to thermal cracking in the initial period of maturation [4,5]. The surface of the concrete mix will cool faster than its core, causing the temperature to rise between the different layers of the structure. Thermal differences between parts of the structure will cause tensile stresses on the concrete surface, and if these stresses are higher than the actual tensile strength of the concrete, then concrete cracking occurs [6]. Thermal cracking will depend on material, structural, and manufacturing factors [7,8].

Practical measurements to mitigate these effects include actions such as changes in the formulation of the concrete mix (e.g., the use of metallurgical cement or a lower cement content), modifications to the structure design (such as additional reinforcement, pre-stressing, expansion joints), or component cooling mixes or installation of cooling pipes or replacement of part of the mixing water with crushed ice. Additionally, the use of PCM with the appropriate phase change temperature may be a good solution in reducing the risk of thermal cracking in young concrete [9,10,11,12].

The current article presents laboratory tests showing the effect of PCM on the maturation process of concrete in slabs with dimensions of 52 × 52 × 20 cm. Temperature changes at different points of the slabs were compared for the case without PCM and with different PCM contents. The results of computer simulations are also presented, taking into account the complexity of the conditions in the thermal chamber during the tests, with particular emphasis on the heat that PCM return and absorb during concrete maturation. The results of laboratory and numerical tests were compared.

## 2. Description of the Experimental Analysis

### 2.1. Materials

In the current research the applicability of PCM is discussed in order to reduce the negative thermal effects in concrete in the initial stage of maturation. PCM are to ensure internal care, preventing the formation of negative thermal gradients inside the manufactured element. Various techniques may be used to add PCM to the concrete mix in order to control temperature changes as it matures. The main methods are the following (see Figure 1):use of PCM pipes in concrete,use of lightweight aggregates impregnated with PCM,adding powder or in liquid PCM microcapsules to the concrete,impregnation of concrete with PCM [13].

There are many possibilities of PCM application in individual fields of science and practice: in construction, transport, agri-food industry, automotive, chemical, medicine, and the textile industry. PCM undergo a phase change within a certain temperature range. When the temperature of the mixture reaches the phase transition temperature, the PCM absorb excess heat, they switch into the liquid, and prevent the temperature from exceeding unfavorable levels inside the concreted element. When the temperature drops below the phase transition temperature, they solidify and give up the excess of stored heat, thanks to which they stabilize the temperature inside the concreted element. The described mechanism of action of PCM in the maturing concrete is favored by typical climatic conditions of the Middle East, characterized by a significant diurnal temperature amplitude [11].

When introducing PCM into concrete, the following requirements should be taken into account. First, the introduced PCM should not interact with the mixture components during the cement hydration process. This can negatively affect the properties of mature concrete. Second, the phase transition period of the PCM should fall within the preferred range for the hydration process. This temperature range is from 19 °C to 34 °C. The last requirement is that the PCM has a high heat capacity. The most frequently used PCM to control the temperature of young concrete is microencapsulated Micronal powder [14,15,16,17,18,19,20,21].

Table 1 lists the individual PCM that can be used in construction, depending on the origin, either organic or inorganic [22]. Currently, all over the world there is a wide variety of PCM, produced by many companies: Rubitherm, Doerken, BASF (Ludwigshafen, Germany), EPS Ltd. (Nottingham, UK), PCM Thermal Solutions (Naperville, IL, USA), Climator (Skövde, Sweden), Cristopia (Vence, France), Mitsubishi Chemical (Tokyo, Japan), TEAP Energy (Australia), PCMS (Shanghai, China), and PlusPolymer (Gurugram, India) [22,23]. In the laboratory experiments, Micronal powder and Rubitherm liquid were used.

### 2.2. Methods

The issues of the use of PCM in construction, with particular use of the temperature of the concrete mix to regulate (shifting in time and lowering the thermal peak) in dry and hot climate conditions, has been described in detail in the doctoral dissertation [11] of the co-author of this article and his numerous publications [1,9]. There are explained the methods of modeling cyclical daily changes in ambient temperature (in proprietary air-conditioning chambers and a chamber of own construction), the influence of various types of PCM on the ability to accumulate and give off heat by a concrete mix, the influence of the type of cements (from high to low calorific values) on the decomposition of temperatures over a period of several days of concrete maturation, the influence of the thickness of the control boards on the uniform heat flow, and the influence of various combinations of these factors on obtaining the most conditions for maturing concrete. The large amount of technological information obtained in this way prompted the authors to undertake work on numerical modeling of the empirically described phenomena, especially as promising results were also obtained in the use of hybrid PCM additives in the powder and liquid versions. Against the background of a successful laboratory experiment, an appropriate numerical model is presented which gives the opportunity to analyze any combination of material and environmental conditions of concrete maturation processes under extreme conditions.

In the below described laboratory experiment, two types of PCM were selected for the research. The first is Micronal, an encapsulated additive, which is in the form of a dry powder from BASF [24]. The second PCM is liquid and comes from Rubitherm [25]. The first one in capsules is programmed to absorb heat when the temperature of the mixture exceeds 23 °C and to release thermal energy when the temperature drops below 26 °C. The second one is liquid and the phase transition temperature ranges from 31 °C to 34 °C. It starts absorbing heat when the temperature exceeds 31 °C, and releases heat to the environment when the temperature drops below 34 °C. Therefore, it is convenient to dose the liquid PCM with the mixing water after mixing the dry concrete components (cement, mineral additives, and aggregates) and, in the final mixing phase after homogenizing the components, to add powder PCM so as not to damage their structure. According to the authors, two-stage dosing guarantees a good distribution of PCM in the volume of concrete.

The accepted recipes for concrete mixes of the C30/37 class and the consistency of S4 are presented in Table 2. As part of the experiment, three slabs (samples) with dimensions of 52 × 52 × 20 cm were made. The first slab did not contain any additions, and was treated as a reference sample, marked as 1^without^ (Table 2). The second and third slabs had an addition of two PCM, abbreviated—2^PCM^ and 3^PCM^ (Table 2). All three samples (slabs) were molded in the laboratory at the same time, with an ambient temperature of around 25 °C.

For research purposes, a modern test setup prepared in the laboratory of the Poznan University of Technology, where experiments can be carried out in climatic conditions similar to those in hot and dry countries. Figure 2 shows a view of three slabs placed in the climatic chamber and a view of the chamber from the outside with a temperature recorder.

The main tests were carried out with the use of a new, specially built climatic chamber, which met the required requirements. Namely, it allowed the adjustment of the ambient temperature in the range (from −50 °C to + 100 °C) and maintained humidity from 0% RH to 100%. Temperature and relative humidity control system in the climatic chamber shown in Figure 2 consist of the following components: humidifier, heater, cooler, and secondary heater. It allowed also the regulation of the temperature 24 times a day in accordance with the assumed daily cycle. Resistance temperature measurement sensors Pt100 were used. They are wire-wound thermoresistors with a measuring accuracy of ± 0.15 °C to ± 0.35 °C. The measurements were performed simultaneously for three samples, setting the course of the daily thermal cycles given in Figure 3, Figure 4 and Figure 5.

It is expected that the excess heat generated in the concrete mix as a result of hydration and the heat from the surroundings during the initial maturation period of young concrete will be absorbed during the melting process of the PCM. Conversely, excessively absorbed heat will be released when the ambient temperature drops below the phase transition temperature of the PCM, i.e., during solidification of the PCM. In this way, the microcapsules are supposed to regulate the temperature of the concrete mix, making the temperature of the maturing concrete uniform and reducing the thermal peak and shifting it in time, as well as enabling the technological barrier associated with quick cement hydration during concreting in dry and hot climates to be exceeded. For the analysis of this phenomenon, cyclic daily cycles of temperature fluctuations (range from 13.6 °C—night to + 43.6 °C—day) were adopted and the response temperatures of two types of samples were recorded.

It should be clearly stated that the conducted laboratory experiment strictly relates to the practice of concreting in dry and hot conditions. The production and transport of concrete in a dry climate takes place in the evening and night conditions, so that the laying of concrete takes place at the lowest possible ambient temperatures, to delay the evaporation of water from the concrete mix. With increasing ambient temperature, with sunrise, we are already beyond the hydration thermal peak of young concrete and the addition of phase change materials additionally lowers the hardening temperature and reduces the shrinkage effect in the structure.

### 2.3. Experimental Results

In the laboratory tests performed, this rule was followed. The concrete mix was prepared at a temperature of about 25 °C and, after shaping three slabs, they were placed in the chamber, which cooled the concrete in the coming hours to simulate the effect of the night chill. The later diurnal cycles simulate—hourly intervals—an increase or decrease in ambient temperature, as occurs in the summer, for example in Damascus.

The ambient temperatures in the chamber were measured. Additionally, the temperatures were measured in three places in each slab (on the bottom and upper surfaces and in the middle of the slab). The initial temperature of the unmodified concrete slab—No. 1^without^ (P1) and the chamber were at a comparable level (see Figure 3). For slabs No. 2^PCM^ (P2) and No. 3^PCM^ (P3), the initial temperature in the chamber was at the same level as for No. 1^without^ (P1), but the initial temperature of both the PCM concrete slabs was lower, because when the phase transition temperature is exceeded, the thermal peak in the concrete is reduced and heat is accumulated by the phase change capsules (see Figure 4 and Figure 5).

Figure 3 shows the temperature course in the slab without PCM (without). When analyzing the first graph, it can be seen that for over 15 h the temperature of the mixture was higher than the temperature of the climate chamber. After 15 h, the slab temperature equalized with the chamber temperature and this condition lasted for 9 consecutive hours. After another 24 h, the temperature in the slab was much higher than the ambient temperature. This condition favors the formation of cracks in the structure [9]. It has been shown that the risk of cracking of the concrete structure in a slab without PCM additives is almost 1.5 times greater than in slabs where PCM have been added.

Figure 4 and Figure 5 show the history of the temperature in second and third slabs to which two PCM have been added. It should be noted that the temperature in the boards with the addition of PCM remains below 24 °C for 18–19 h, which significantly reduces the possibility of cracks in this critical period. The temperature in the concrete mix is also kept below the temperature in the climatic chamber for the first 24 h after its production. This condition creates almost perfect conditions for the maturation of concrete.

The temperature curve (second slab) quickly lowers after reaching the maximum, which proves that PCM quickly release the absorbed heat to the environment. This feature is very important for the durability of the structure, not only for the structure made in hot climates, but also for the structure erected in Polish climate, e.g., a massive structure. Figure 6 shows a comparison of the temperature course in the middle of the three slabs. The temperature difference between the slab without PCM and the slabs with the addition of two PCM initially ranges from 8 °C to 5 °C in the later hours of concrete maturation, which confirms the effectiveness of using such materials in the elements made in hot climate regions.

## 3. Computer Simulations of the Process

The experimental studies using the previously described test stand (Figure 2), which was used to ensure a suitable daily cyclically variable temperature of concrete maturation, have been extended by computer simulations. In the experimental tests, temperature distribution was obtained during the first 3 days from the beginning of the maturation process. Modeling of hydro-thermo-mechanical processes occurring during concrete maturation has been widely described in the literature [26,27,28,29,30,31,32]. In numerous examples, mathematical models have been verified and material parameters have been identified. Such tests have led to a situation in which it is possible to predict the behavior of concrete, including its strength, and even the mechanism of destruction of the concrete structure, along with the correct determination of its load-bearing capacity, based on computer simulations, taking into account hydro-thermo-mechanical conditions. In the computer simulations presented in this work, the authors focused only on the possibility of modeling changes in the temperature field, taking into account additives in the form of PCM. The PCM task is to lower the average concrete maturation temperature by changing their aggregate state, during which they absorb or release a certain portion of thermal energy. Reducing the maximum temperature reached by the concrete during maturation reduces the formation of internal cracks.

### 3.1. Transient Heat Flow

The energy balance of the system according to the first law of thermodynamics can be represented by the following equation [33,34,35]:(1)$$\int_{V}\rho \dot{U}dV=\int_{S}qdS+\int_{V}rdV,$$
where V is the volume of the material with the external surface S. In the equation, the material density is also denoted as ρ, $\dot{U}$ is the rate of changes of the internal energy U, q is the heat flux on the outer surface of the body under consideration, and r is the heat generated inside the material as a result of, for example, chemical reactions. As already mentioned, only temperature changes will be considered, and therefore the internal energy depends only on temperature T and time t, which is represented by the relationship U=U (T, t). To solve the above equation for specific thermal conditions, the Abaqus/Standard program was used, which takes into account both the heat flow inside the material as well as the inflow and outflow of heat to and from the surface as well as radiation.

Disregarding the influence of the displacement field, the constitutive law of the material determines the specific heat c as a function of temperature T according to the following equation:(2)$$c\left(T\right)=\frac{dU}{dT}.$$

The rate of change of internal energy $\dot{U}$ can be determined based on the following equation:(3)$$\dot{U}=\frac{dU}{dT}\dot{T}=c\left(T\right)\dot{T}.$$

Typically, it is assumed that the latent heat L is an addition to the specific heat c, which appears or disappears between the lower $T_{S}$ and the higher temperature $T_{L}$ of the phase transition. dU/dT is the specific heat c(T) outside the phase transition and is equal to $c+L/\left(T_{L}-T_{S}\right)$ in the phase transition when $T_{L}>T>T_{S}$. In this respect, the internal energy is influenced by the latent heat L (Figure 7 and Figure 8).

In many cases it makes sense to assume that the phase transition occurs in the temperature range known for a given material. This situation also occurs in the case of PCM, the behavior and influence of which on the concrete maturation process is discussed in this paper. Such an approach without taking into account the kinematics of the phase transition process is simplified and in some cases not sufficient [33,35].

The thermal conductivity of the material can be described by Fourier’s law [35,36]. The heat flux f is calculated using the following equation, taking into account the temperature gradient ∂T/∂x:(4)$$f=-k\frac{\partial T}{\partial x},$$
where k is the coefficient of thermal conductivity generally dependent on the temperature k(T), and x is the position inside the body under consideration. On the external surface of the body under consideration, the boundary conditions should be determined by means of specific temperature changes: T=T(x,t) or by means of the heat flux to the surface q=q(x,t) or volume r=r(x,t). Convection and radiation of heat from the surface of the body $S_{q}$, in turn, can be described by the following relationships:(5)$$q=h\left(T-T_{0}\right)-\mathrm{convection},\;q=A\left({\left(T-T_{0}\right)}^{4}-{\left(T_{0}-T_{Z}\right)}^{4}\right)-\mathrm{radiation}.$$

In Equation (5), h=h(x,t) is the heat film coefficient, $T_{0}=T_{0}\left(x,t\right)$ is the ambient temperature, and A is the radiation coefficient (the product of emissivity and the Stefan-Boltzmann constant).

In the finite element method (the Galerkin approximation), the following interpolations of the temperature and its rate inside the finite elements are used:(6)$$T=N^{N}\left(x\right)\;T^{N}\;\mathrm{for}\;N=1,2,\ldots \;\dot{T}=N^{N}\left(x\right)\;{\dot{T}}^{N}\;\mathrm{for}\;N=1,2,\ldots$$

In this equation $N^{N}$ are matrices with shape functions that are usually first or second degree polynomials in a one, two, or three dimensional space. $T^{N}$ is the vector of the sought nodal temperatures of the finite element. Then the system of equations for a single finite element takes the following form:(7)$$\int_{V}N^{N}\rho cN^{M}dV\;{\dot{T}}^{M}+\int_{V}\frac{\partial N^{N}}{\partial x}k\frac{\partial N^{M}}{\partial x}dV\;T^{M}=\int_{S_{q}}N^{N}qdS+\int_{V}N^{N}rdV,$$
assuming the discussed spatial discretization by means of Finite Element Method (FEM). The first term in Equation (7) is close to zero under steady heat flow conditions. In the case of a transient flow, Equation (7) is integrated over time. In matrix form, the system of equations can be written in the following form:(8)$$C\;\dot{T}+K\;T=Q,$$
where:(9)$$C=\int_{V}N^{N}\rho cN^{M}dV,\;K=\int_{V}\frac{\partial N^{N}}{\partial x}k\frac{\partial N^{M}}{\partial x}dV,\;Q=\int_{S_{q}}N^{N}qdS+\int_{V}N^{N}rdV.$$

In Equation (8), C is the heat capacity matrix, K is the thermal conductivity matrix, Q is the external and internal heat flux vector, and T is the node temperature vector.

In the discussed case of transient heat flow, the following backward differential algorithm was used:(10)$${\dot{U}}_{t+\Delta t}=\left(U_{t+\Delta t}-U_{t}\right)\left(\frac{1}{\Delta t}\right).$$

We write the heat flow equation for the moment t+Δt assuming that the residual heat flux $R^{N}$ at each node is equal to zero:(11)$$R^{N}\left(T_{t+\Delta t}^{M}\right)=\frac{1}{\Delta t}\int_{V}N^{N}\rho \left(U_{t+\Delta t}-U_{t}\right)dV\;+\int_{V}\frac{\partial N^{N}}{\partial x}k_{t+\Delta t}\frac{\partial N^{M}}{\partial x}dV\;T_{t+\Delta t}^{M}-\int_{S_{q}}N^{N}q_{t+\Delta t}dS-\int_{V}N^{N}r_{t+\Delta t}dV=0.$$

Assuming that $c_{i+1}^{M}$ is a correction of the approximate solution $T_{t+\Delta t,\;i}^{M}$, we obtain the following equation:(12)$$R^{N}\left(T_{t+\Delta t,i}^{M}+c_{i+1}^{M}\right)=0.$$

When solving the above equation using the modified Newton’s method, it should be decomposed into the Taylor series around the approximate solution $T_{t+\Delta t,\;i}^{M}$, which leads to:(13)$$R^{N}+\frac{\partial R^{N}}{\partial T^{M}}c_{i+1}^{M}+\ldots =0,$$
where $\partial R^{N}/\partial T^{M}$ is the Jacobian matrix (tangential stiffness) K. Iterations are carried out until $R^{N}$ and $c_{i+1}^{M}$ are sufficiently small.

### 3.2. Numerical Modelling of Laboratory Tests, Parametric Study

In order to determine the extent to which all the parameters of the simulation (constitutive law, boundary conditions, etc.) affect the change and temperature distribution in the concrete slab, computer simulations of the process were carried out in the Abaqus/Standard program. In this work, the focus was only on the simulation of thermal problems and the part related to the strength (displacement) analysis in the potential sequential temperature-displacement analysis was omitted (Figure 9).

The regular geometry of the slab (dimensions 0.52 m × 0.52 m × 0.2 m) allows the use of rectangular finite elements (diffusive heat transfer elements). The use of linear elements with the designation: DC3D8 (8-node linear heat transfer brick) and parabolic ones with the designation: DC3D20 (A 20-node quadratic heat transfer brick) was considered. In all kinds of computer simulations, the first point should be to determine the effect of discretization on the results obtained. The influence of the element type along with the mesh density was also analyzed in the current study. The ambient temperature was assumed to be the temperature set by the programmable controller and then measured in the thermal chamber, which is assumed during the experiment to determine the rhythm of daily temperature changes in Middle East (Syria), Figure 10. The ambient temperature during the tests varied between 13.8 °C and 43.3 °C with an average value of 27.31 °C. The model takes this fact into account as a convection phenomenon in which the heat flux q has been determined assuming the appropriate inflow and outflow coefficient h (constant during the simulation) and the change in ambient temperature known from the experiment. The influence of its value on the results, i.e., temperature changes at the measurement points, was determined (Figure 9). Convection is represented in Equation (7) as the third integral.

The second analyzed parameter is the coefficient of thermal conductivity k. It determines the speed of thermal wave propagation in the form of heat flux f inside the body under consideration. In Equation (7) this fact is included in the form of the second integral. The third analyzed thermal parameter is the specific heat c. It determines the heat capacity of the material. In Equation (7) this fact is taken into account by the first integral. An important element of the analysis is the additional heat flux r. It is heat generated, for example, by a chemical reaction related to the maturation of concrete. In the analyzed problem, there is also latent heat related to the phase change. PCM absorb and release heat depending on the temperature. In Equation (7) this fact is taken into account in the form of the fourth integral. In order to take into account both mechanisms of the formation of the internal heat flux r, the HETVAL user procedure was used. To sum up, the impact on the history of temperature changes at the measurement points of the following parameters will be determined in computer simulations: h, k, c, and flux r (the HETVAL procedure, which takes into account latent heat of the phase transition). In the parametric study, the initial temperature of the concrete was assumed to be 27.77 °C, i.e., the one that was in the concrete without PCM during the experiment when it was placed into the thermal chamber. The values of all the material parameters are presented in Table 3. In all numerical simulations the SI unit system was used. It means that the unit of time was second [s]. These units are also visible on all the curves which present purely numerical results. However, to clarify, 100,000 s is about 27.7 h.

#### 3.2.1. Discretization

For discretization purposes, material constants ρ, k, and c were adopted as shown in Table 3. Finite elements were adopted to describe the heat flow with linear shape functions DC3D8 with dimensions of 0.08 m, 0.04 m, and 0.02 m and with parabolic shape functions DC3D20.

The temperature distributions are presented in Figure 11 (cooling and heating according to Figure 10) for linear shape functions and different finite element sizes but similar distribution is visible in case of quadratic shape function. The results in the meaning of the temperature are comparable. Additionally, it is also visible that the corners of the slab have higher difference between the highest $T_{max}$ and lowest $T_{min}$ temperatures—in further experiments the temperature should also be measured in this area. The temperature difference between middle and top is observed due to conductivity k, which describes heat flux.

The conducted analyses showed that in this case the size of the finite elements and their type are not of great importance (Figure 11). In further calculations, the finest mesh of 0.02 m from the DC3D8 elements was used. However, the results were mesh size insensitive.

#### 3.2.2. Conductivity

The influence of thermal conductivity was analyzed in the study because it determines the rate of temperature changes inside the concrete slab. The coefficient k in general depends on the temperature, but in this range, which occurs in the discussed example, it is practically constant. The two values of k were taken into account: 0.7 and 1.7 J/(s m ℃) (Table 3 and [33]). Such values were measured for various types of concrete, so they are real possible values of this parameter. In this study, these values were not measured directly [33]. Figure 12 shows the influence on the results, in the form of temperature changes at selected points inside the slab for both thermal conductivity coefficient k.

For the smaller k, a greater temperature difference is seen between the upper and middle points of the slab at the moments of the lowest (b) and highest (a) ambient temperature $T_{0}$ (temperature in the thermal chamber in the experiment, Figure 10). This is because there is less heat flux inside the slab for smaller k and the temperature therefore equalizes more slowly. This effect is also visible in experimental studies. As a result of the reduction of the thermal conductivity coefficient, the maximum temperature difference in the concrete also decreased (for k equal to 1.7 J/(s m ℃), the maximum temperature difference ΔT was 14.9 ℃ and for k equal to 0.7 J/(s m ℃) it was 11.7 ℃).

#### 3.2.3. Film Condition

The convection phenomenon in which the heat flux q was determined assumed the appropriate inflow and outflow coefficient h and the change in the ambient temperature known from the experiment. The parameter h was assumed to be constant during simulation. It is true in this range of temperatures.

The influence of its value on the results was determined, i.e., temperature changes at measurement points for two values of the coefficient h, i.e., 10 $J/\left({s\;m}^{2}\;℃\right)$ and 5 $J/\left({s\;m}^{2}\;℃\right)$, Table 3. In the case of a smaller h coefficient, the difference between the lowest and the highest temperature at the measuring points was 10 ℃ compared to 14.9 ℃ for the coefficient with a larger value, Figure 13.

#### 3.2.4. Specific Heat

The influence of specific heat c, which determines the heat capacity of the material, on the temperature change in the concrete slab, is shown in Figure 14. Two cases were compared where the specific heat is equal to 880 J/(kg ℃) and 500 J/(kg ℃), see Table 3.

The presented results show that for a lower specific heat c, equal to 500 J/(kg ℃), the temperature increase inside the slab is faster and in the simulations a higher maximum temperature difference ΔT of 20.5 ℃ was also noted (instead of 14.9 ℃ for the specific heat equal to 880 J/(kg ℃)).

#### 3.2.5. Heat Generation

In maturing concrete, heat is generated by an exothermic hydration reaction. As a result, the temperature of the concrete rises. In addition, thermal energy is absorbed and released by the PCM, which reduces the temperature inside the concrete slab. The general model was assumed in the following additive form:(14)$$r\left(T,t\right)=\alpha_{C}r_{H}\left(t\right)+r_{L}\left(T\right),$$
where $r_{H}\left(t\right)$ is the time-varying heat flux associated with cement hydration, where $\alpha_{C}$ is a parameter related to the percentage by volume of cement in the concrete mix, while $r_{L}\left(T\right)$ is the heat flux associated with the phase transformation of PCM and latent heat dependent on temperature. In the presented research, two types of PCM with different ranges of operation and different latent heat were used. In order to implement the first part, the volumetric heat flux was used, while in the case of the second part, the HETVAL user procedure was used, which will be presented later in this work. In the case of cement, the hydration reaction generates the following amount of heat, Figure 15.

The history of temperature changes corresponding to the $r_{H}\left(t\right)$ flux is shown in Figure 16. It presents a numerical solution using the modified Euler method of the following differential equation:(15)$$\frac{dT}{dt}=\alpha_{C}\frac{r_{H}\left(t\right)}{\rho \;c},$$
according to the following formula:(16)$$T_{n+1}=T_{n}+\alpha_{C}\frac{\Delta t}{\rho \;c}\frac{\left(r_{H,n+1}+r_{H,n}\right)}{2},$$
for $T_{0}=27.77\;℃$. It also includes a solution for simulation in Abaqus. It was observed, in accordance with [38], that the maximum temperature increase was equal to about 21.73 ℃ (Figure 16).

It should be noted that the experimental curves [37,38] deal with cement. In concrete, only a part of the volume is responsible for the heat increase, which can be taken into account in the model by introducing the parameter $\alpha_{C}=0.14$. It is clear that the temperature cannot drop when there is a positive volumetric heat flux $r_{H}\left(t\right)$ (Figure 15). However, the temperature of the mortar in the calorimeter decreased most likely due to heat loss at the edge. Taking this fact into account, two sets of parameters were calibrated—assuming the ideal insulation of the calorimeter ($\alpha_{C}=0.1$) and its absence ($\alpha_{C}=0.14$), and assuming the heat inflow coefficient $h=0.25\;J/\left({s\;m}^{2}\;℃\right)$ and ambient temperature $T_{0}=20\;℃$. The results of the computer simulation for the presented model (cube 0.1 m × 0.1 m × 0.1 m—64 finite elements) and the solution of the differential equation based on the modified Euler’s method are shown in Figure 16.

Another analyzed issue is the absorption and emission of heat by the phase change material. The two different PCM have different latent heat and a different transformation temperature. Figure 17 shows the model proposed in this paper that takes this fact into account for the two PCM used in the study.

The heat flux associated with the change of the physical state $r_{L}\left(T\right)$ of both materials was assumed in the following form:(17)$$r_{L}\left(T\right)=L_{i}\left(T\right)=a_{i}T^{2}+b_{i}T+c_{i}\;\;\;\;\;\mathrm{dla}\;\;\;T_{Si}\leq T\leq T_{Li}.$$

In this equation, the parameters $a_{i}$, $b_{i}$, and $c_{i}$ were determined on the basis of the course of the functions shown in Figure 17. The parameter values corresponding to this function are presented in Table 4 (for *i* = 1, heat is released but for *i* = 2 and 3 it is absorbed).

HETVAL user subroutine has been programmed to account for the heat generated by the phase change. The procedure interface is as follows (Simulia, 2019) (Table 5).

The following variables are defined in the subroutine: FLUX(1) is heat flux, $r_{L}$—thermal energy per time per volume: $J/\left({s\;m}^{3}\right)$ at the material calculation point, FLUX(2) is the rate of change of heat flux per temperature $\partial r_{L}/\partial T$. The additional variables which can be defined are STATEV(*). This is an array containing the user-defined solution-dependent state variables at this point. The variables need to be specified using **Depvar* function and later written into Abaqus output files as SDV variables. CMNAME is the material name, TEMP(1) is the current temperature, and T and TEMP(2) is temperature increment, ΔT. The step time at the end of the increment is TIME(1) and TIME(2) is total time. Additionally, DTIME is time increment Δt. An array containing the initial values at the beginning of the analysis and the current values during the analysis of the user-specified field variables are defined as PREDEF(*) and DPRED(*) is an array containing increments of these variables. The subroutine used in current simulation is presented in Appendix A. The rate of change of heat flux per temperature is calculated using the following equation:(18)$$\frac{\partial r_{L}}{\partial T}=\frac{\partial L_{i}\left(T\right)}{\partial T}=2a_{i}T+b_{i}\;\mathrm{dla}\;\;\;T_{Si}\leq T\leq T_{Li}.$$

The parameters are presented in Table 4.

## 4. Comparison of Experimental Results and Computer Simulations

The above analysis of the influence of the model parameters, those that determine the behavior of the material and those that determine the boundary conditions and the interaction of the concrete slab with the environment (Table 3 and Table 4), led to the creation of two computer models. The first concerned the propagation of heat in the slab without PCM, and the second with PCM. The experimental results presented in Figure 18 have shown a reduction and equalization of temperature in slab with PCM (dash lines: blue—without PCM and grey—with PCM). The same effect was obtained in the simulations (see Figure 18 (full lines: orange—without PCM and yellow—with PCM)). The temperature in the middle of the slab decreased by an average of 3 °C.

The following figures compare the temperatures in the slabs with and without PCM during concrete maturation (see Figure 19) at different time points, (a) after 12 h, (b) after 25 h, and (c) after 40 h. In each case on the left there is temperature distribution for slab without PCM, in the middle with PCM. The right side shows that in each case the plot is similar to the one presented in Figure 18, marked with a yellow line corresponding to the appropriate time. The presented analysis leads to the final models and simulations which correctly describe the temperature condition in the concrete slabs during cyclic heating and cooling periods (day/night conditions).

The important aspect is also the fact that PCM start to work at the beginning before we place the specimen in the thermal chamber. Thanks to this the initial temperature of the concrete mix is about 8 °C lower in case of the slab with PCM.

## 5. Conclusions

In hot and dry climate conditions, it is justified to use technological flexibility in the form of several maintenance options depending on the conditions of concreting and maturing, technological possibilities of the plant, and the type of concrete element. PCM used for the modification of cement concrete make it possible to exceed typical technological barriers in the case of concreting in unfavorable conditions (hot and dry climate). The use of PCM as a cement concrete modifier makes it possible to limit the intensity of the thermal peak and to standardize the temperature in the cross-section of the concrete element, which reduces the risk of element cracking associated with it (which may disqualify it). The addition of PCM to concrete also reduces the magnitude of temperature changes during the operation of the concrete element, which leads to increased durability and an extension of the life cycle of such element. The addition of two PCM directly to the maturing concrete delayed the appearance of the thermal peak (which was finally lower), which allowed the construction of a correct concrete structure resistant to cracks and scratches. Thanks to PCM, the initial (starting) temperature of the concrete mix is 8 °C lower than in the slab without the addition of PCM, which helps to build the correct structure of the young concrete. The use of PCM saves a huge portion of energy without any significant reduction in mechanical strength [39].

The research work considered the possibility of adding PCM to concretes with both high-caloric and low-caloric cements. Of course, only adding PCM to concretes with high-calorific cements is effective, because then the thermal peak in the concrete is lowered by a few degrees °C, and, furthermore, it is shifted in time by several hours (concreting in the sun, setting processes after sunset), which has a direct impact on contraction and bleeding phenomena. In practice, for the construction of certain structures (bridges, concrete highways) mainly high-calorific cements are used due to the guarantee of maintaining the subsequent mechanical properties. PCM is preferred as a concrete curing method, because other curing methods associated with lowering the curing temperature of concrete are simply more expensive.

The tests carried out so far confirm that the applied powdered and liquid PCM are characterized by the fact that when the PCM reaches the phase change temperature, by absorbing the excess heat, they do not allow the unfavorable temperature level inside the concrete element to be exceeded. When the temperature drops below the phase transition temperature, the PCM solidify and give up the excess stored heat, stabilizing the temperature inside the concrete element. The described mechanism of action of PCM in the maturing concrete is favored by the climatic conditions of the Middle East, characterized by a significant diurnal temperature amplitude. The heat accumulated during a hot day in concrete walls of residential buildings with the addition of PCM is transferred during a cold night, thus adjusting the thermal comfort to the needs of residents [40]. In order to design the proper thermal efficiency of such walls, numerical simulations [41] are extremely useful, allowing for the adjustment of the thickness of the partitions to the required parameters, which create a room heat balance favorable for humans.

The presented calculations have shown that supporting experimental research with calculations can help in understanding the processes taking place, including in the case of the use of PCM. The results obtained in the simulations reflect the qualitative and quantitative results of the experimental research surprisingly well. This aspect was the primary goal of the research.

## Figures and Tables

**Figure 1 materials-15-04687-f001:**
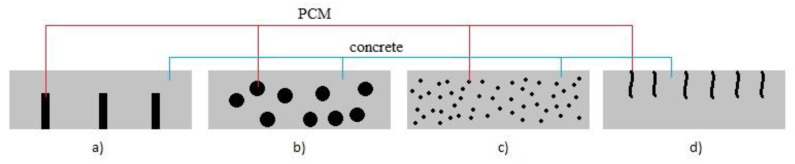
Methods for PCM incorporation in concrete: (**a**) using pipes filled with PCM; (**b**) using lightweight aggregate particles impregnated with PCM; (**c**) using microcapsules with PCM; (**d**) filling concrete surface voids via PCM absorption (adapted form [13]).

**Figure 2 materials-15-04687-f002:**
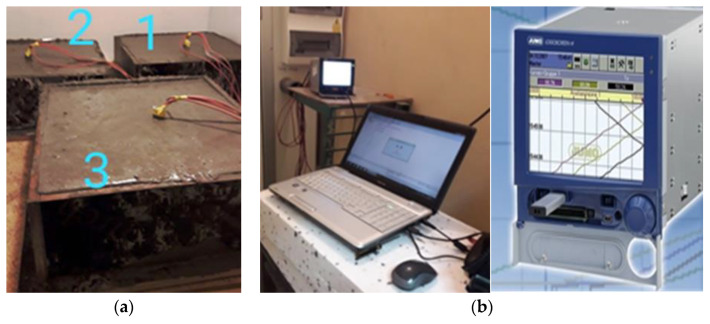
View of the concrete slabs without PCM (1) and with PCM (2,3) in the climatic chamber (**a**) and computer measurements (**b**).

**Figure 3 materials-15-04687-f003:**
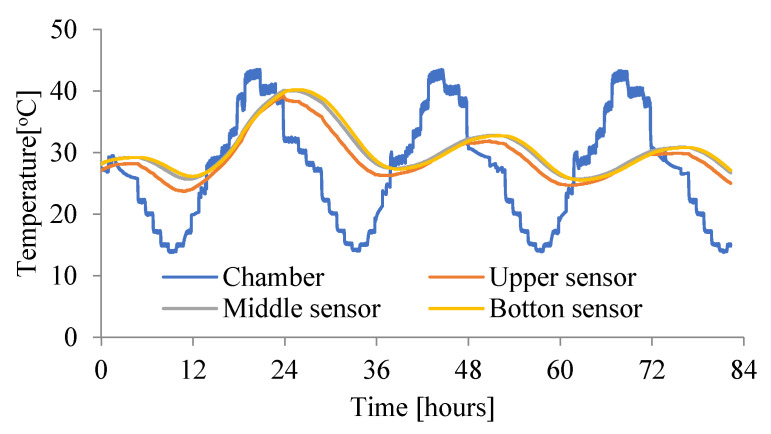
Change of the temperature in slab—No. 1^without^ (P1—the temperature of the unmodified concrete mix and the chamber are at a comparable level).

**Figure 4 materials-15-04687-f004:**
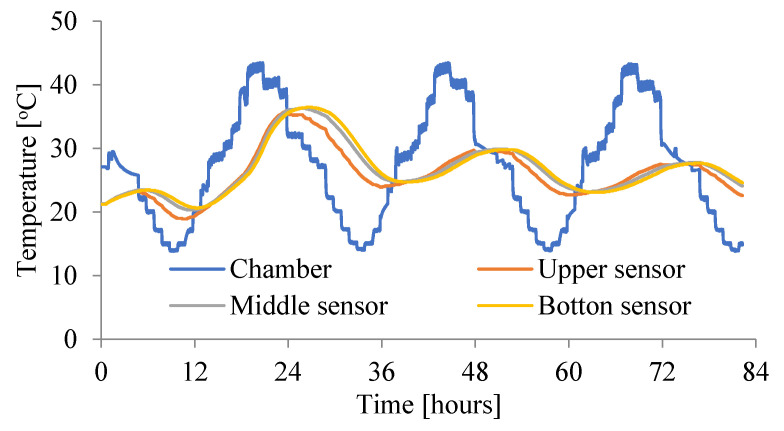
Change of the temperature in slab—No. 2^PCM^ (P2—the temperature of PCM concrete mix is lower than the temperature of the chamber).

**Figure 5 materials-15-04687-f005:**
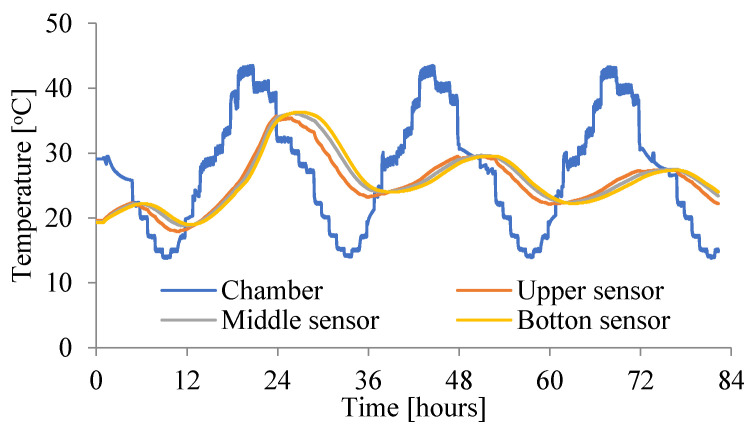
Change of the temperature in slab—No. 3^PCM^ (P3—the temperature of PCM concrete mix is lower than the temperature of the chamber).

**Figure 6 materials-15-04687-f006:**
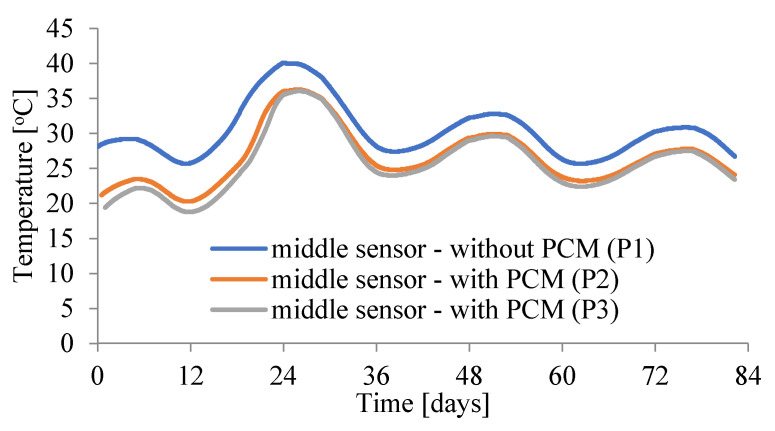
Comparison of the temperature’s changes in three slabs.

**Figure 7 materials-15-04687-f007:**
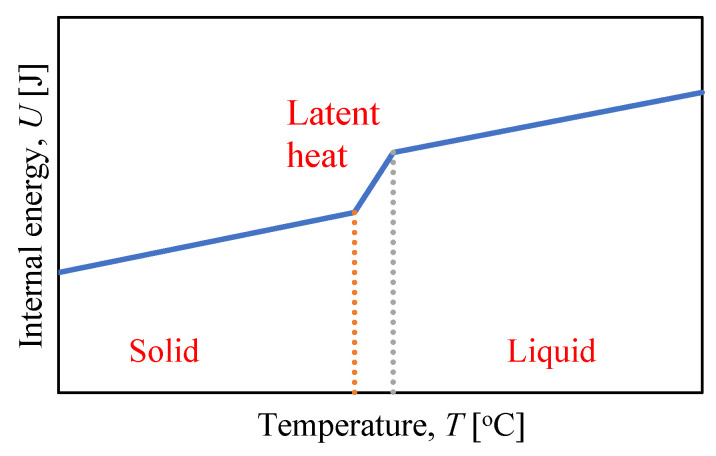
Internal energy versus temperature—specific heat definition.

**Figure 8 materials-15-04687-f008:**
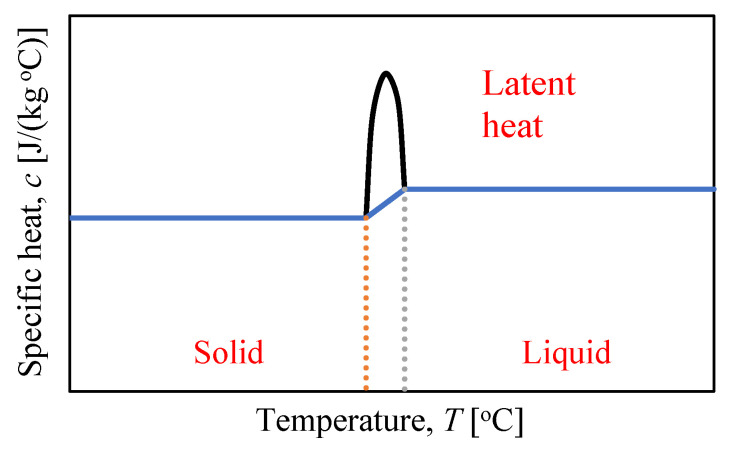
Specific heat versus temperature—latent heat.

**Figure 9 materials-15-04687-f009:**
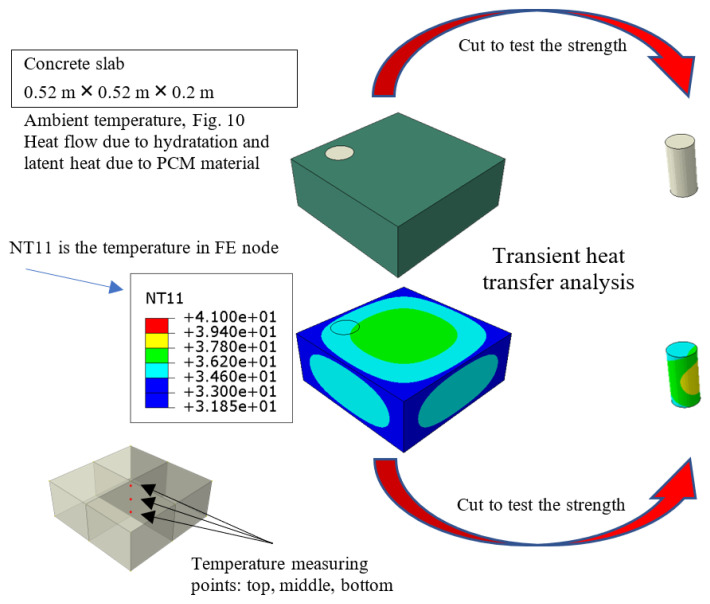
Explanation of the model in potential sequential temperature-displacement analysis.

**Figure 10 materials-15-04687-f010:**
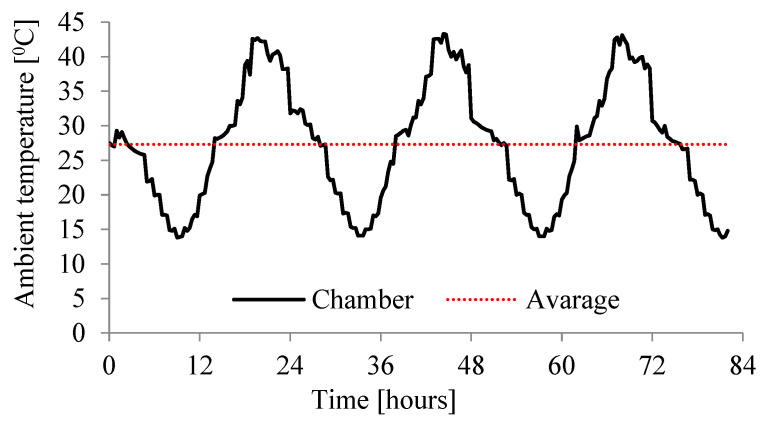
Change in ambient temperature over time (average 27.31 °C).

**Figure 11 materials-15-04687-f011:**
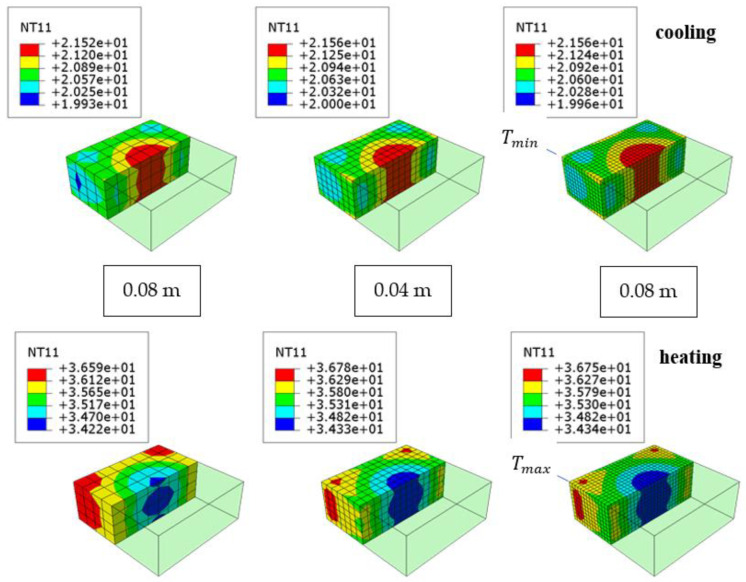
The effect of the element size on the temperature distribution in the concrete slab.

**Figure 12 materials-15-04687-f012:**
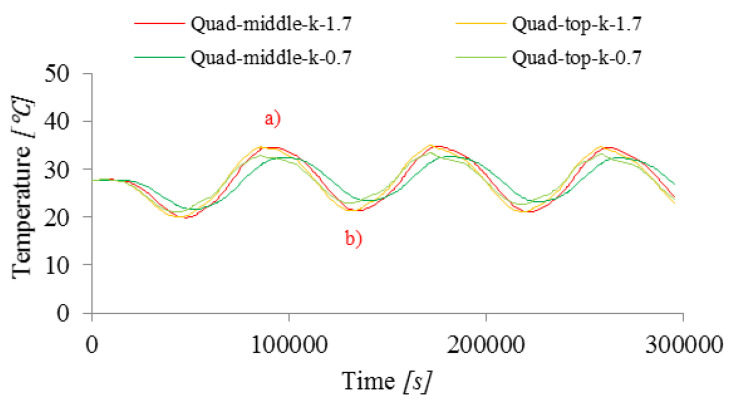
Influence of the conductivity on the change in temperature in time at two measuring points (top and middle)—heating in point a) and cooling in point b).

**Figure 13 materials-15-04687-f013:**
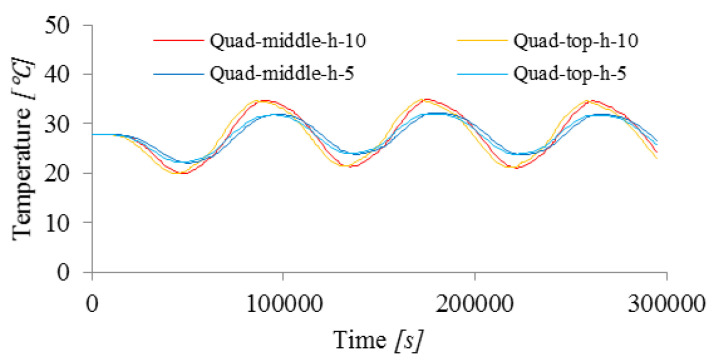
Influence of the film condition on the change in temperature in time at two measuring points (top and middle).

**Figure 14 materials-15-04687-f014:**
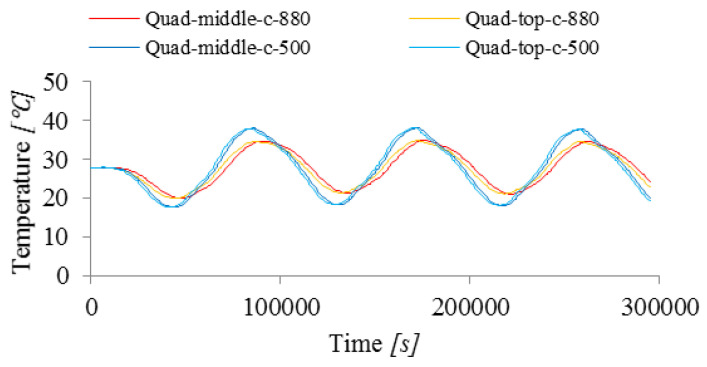
Influence of the specific heat on the change in temperature over time at two measuring points (top and middle).

**Figure 15 materials-15-04687-f015:**
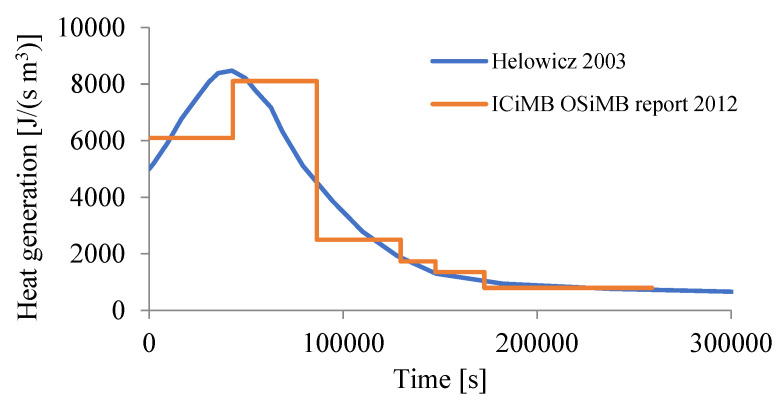
The heat generated due to hydration $r_{H}\left(t\right)$ of cement CEM I 42.5 N (based on [37,38]).

**Figure 16 materials-15-04687-f016:**
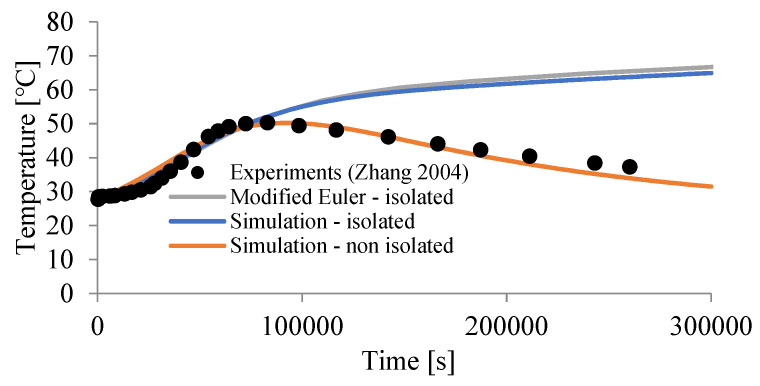
Temperature changes in concrete due to hydration ($T_{0}=27.77\;℃$).

**Figure 17 materials-15-04687-f017:**
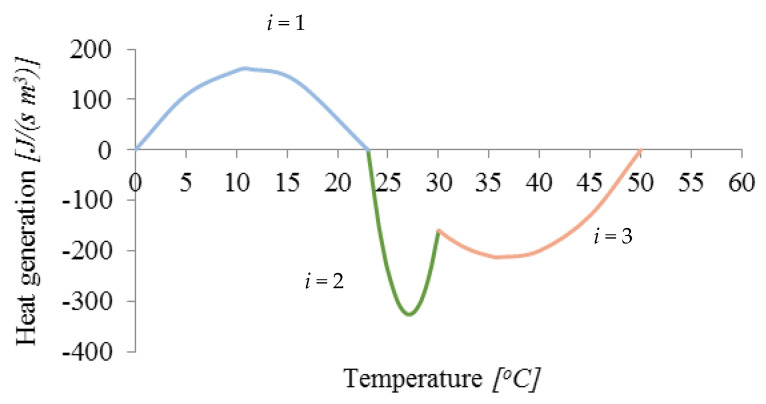
The heat generated due to phase change $r_{L}\left(T\right)$.

**Figure 18 materials-15-04687-f018:**
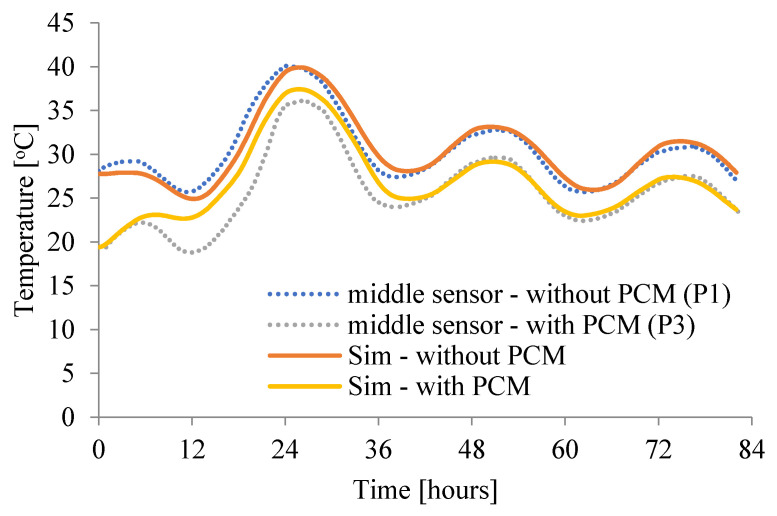
The comparison of the temperature changes in middle of the concrete slab in both experimental and numerical results in two considered concrete slabs (with and without PCM).

**Figure 19 materials-15-04687-f019:**
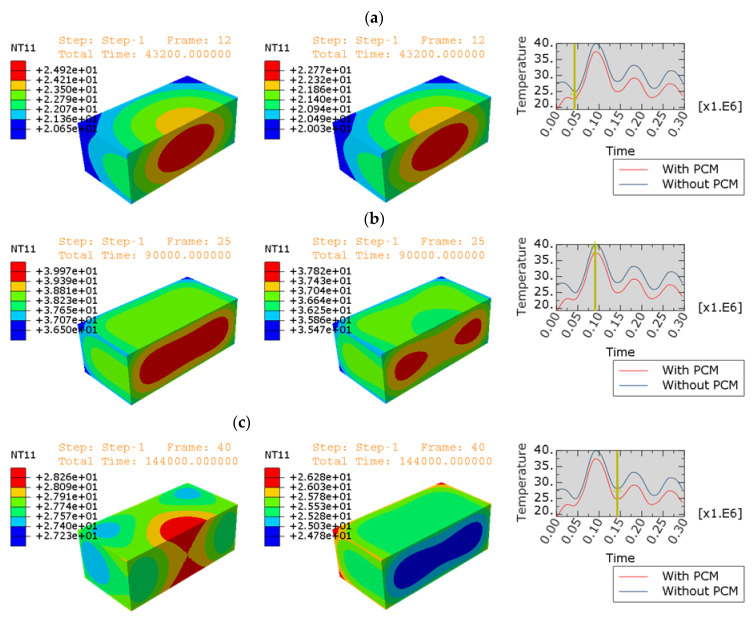
Distribution of the temperature in the concrete slabs (**left**—without PCM and **right**—with PCM) together with variation of the temperature in the middle: (**a**) time=43,200 s=12 h; (**b**) time=90,000 s=25 h; (**c**) time=144,000 s=40 h.

**Table 1 materials-15-04687-t001:** List of physical properties of PCM used in construction and in the current laboratory experiments.

Group	Melting Point Range [°C]	Heat Conductivity [W/m K]	Density Range [kg/m^3^]	Specific Heat Range [kJ/kg]
PCM organic	19–34	0.3	400–900	120–180
PCM inorganic	25–35	0.6	800–1300	120–180
Micronal powder	23–26	0.13	400	142
Rubitherm (liqid 40 °C solid 26 °C)	31–34	0.2	770 880	240

**Table 2 materials-15-04687-t002:** Concrete mix recipes—No. 1^without^ (P1), No. 2^PCM^ (P2), and No. 3^PCM^ (P3) [24,25].

Kind of Materials	Producer	Volume of Materials for 1 m^3^ of Concrete, kg	Density kg/m^3^
Cement CEM I 42,5N	Factory Warta	345	3100
Sand 0/2	Dabrowa	900	2650
Gravel 2/8	Dabrowa	450	2650
Gravel 8/16	Dabrowa	450	2650
Plasticiser BV	SIKA	2.06	1150
Water	Tap water	207	1000
Micronal powder (No. 2^PCM^, No. 3^PCM^)	BASF	36	300–400
PCM liquid (No. 2^PCM^, No. 3^PCM^)	Rubithern	10.35	770–880

**Table 3 materials-15-04687-t003:** Material constants extended by the simulation parameters.

Material Constants	Value
ρ, density [$\left(\mathrm{kg}/m^{3}\right)$]	2400
k, conductivity [J/(s m ℃)]	1.7
c, specific heat [J/(kg ℃)]	880
h, film coefficient [$J/\left({s\;m}^{2}\;℃\right)$]	10
$T_{0}$, sink (ambient) temperature [℃]	Figure 10
k, conductivity [J/(s m ℃)]	0.7
h, film coefficient [$J/\left({s\;m}^{2}\;℃\right)$]	5
c, specific heat [J/(kg ℃)]	500
$r_{H}$, heat of hydratation $\left[J/\left({s\;m}^{3}\right)\right]$	point 3.2.5.
$r_{L}$, heat of phase change $\left[J/\left({s\;m}^{3}\right)\right]$	point 3.2.5.

**Table 4 materials-15-04687-t004:** The parameters of the phase change model.

i	$T_{Si}$ [℃]	$T_{Li}$ [℃]	$a_{i}$	$b_{i}$	$c_{i}$	Heat Flux
1	0	23	−1.20983	27.82609	0	Release
2	23	30	44.08163	−2359.18	30,942.04	Absorb
3	30	50	1.2	−88	1400	Absorb

**Table 5 materials-15-04687-t005:** Interface of the HETVAL subroutine.

SUBROUTINE HETVAL(CMNAME,TEMP,TIME,DTIME,STATEV,FLUX, 1 PREDEF,DPRED) INCLUDE ‘ABA_PARAM.INC’ CHARACTER*80 CMNAME DIMENSION TEMP(2),STATEV(*),PREDEF(*),TIME(2),FLUX(2), 1 DPRED(*) *user coding to define* FLUX *and update* STATEV RETURN END

## Data Availability

Data sharing is not applicable for this article.

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
