# Peer review of "Experimental and Numerical Analysis of the Concrete Maturation Process with Additive of Phase Change Materials"

_materials, 2022, doi:10.3390/ma15134687_

Round 1
Reviewer 1 Report
The paper is clearly presented. Sufficient quality of theoretical background, numerical implementation and calculations.
Author Response
Thank You very much for all Your comments. The answers are in the file.

Reviewer 2 Report
This is a comparative analysis between the results of the laboratory tests and numerical simulations. the paper is within the scope of this journal and can be accepted after some corrections.
- Abstract: This is a general abstract and the authors must add some results in this section
- Keywords: please add one more extra keywords
- The first paragraph in the introduction contains only one reference, I think this is not enough and I encourage the authors to add more Refs
- Introduction: In the last paragraph, the authors must summarize what they did in this work
- 1. Adopted procedures of dosing phase change materials to concrete....why this takes number (1)
- the same comment for 1. Methods for PCM incorporation in concrete
- I think you can write [15, 16, 17, 18, 19, 20,21] as [15-21]
- Please check the numbering for the sections (see 1. Laboratory experiments, 1. Experimental results, 1. Computer simulations of the process)
- Please add number for the equations, all the equations are written without number
- Is there any reason for the difference between the simulation and the experimental results
- In the conclusion, I think you must merge it in one pargraph no need to add different subsections
- Some of the Refs are old, I the authors must update some of the Refs and add recent Refs (2020, 2021 and 2022)
- Please check Ref 37, this is not acceptable
Author Response
Thank You for your thorough review of the article. Our answers are in the file.

Reviewer 3 Report
English is rather poor and requires to be improved and probably by a English native. There is no need to add" material" behind "PCM". Many words are inaccurate (witness instead of reference sample, powder for micronal, etc...)
The organisation of the manuscript must be improved. The introduction must be completed and a little bit developed. the introduction must further contextualize the subject and must present at the end the purpose of this study.
There is almost nothing about the PCMs used in this study. The properties of the 2 different PCMs must be presented in a table (composition e.g. paraffin for micronal, Temperatures of phase change, Latent heat, etc...). Does table deal with the 2 used PCMs?
In a general way, the figure captions are insufficient and must be developed.
The design and manufacturing of the samples must be clearly described. Express the amount of PCM in %. When are the PCMs mixed with concrete? Before hydration or at the end after mixing the main concrete compounds? Figure 2 shows instrumented samples (thermocouples I guess). What kind of thermocouples?
Samples undergo some thermal cycling in a climatic chamber. precise the model of the climatic chamber, relative humidity, accuracy of the thermal monitoring. Do the 3 samples undergo thermal cycling at the same time or different experiments for each sample composition? The experimental process is not sufficiently presented.
Express fig 3 to 6 in hours instead of days.
Why the slab size for numerical modelling is different than experimental study?
What is NT11? Is is the hundredth of a degree relevant to express the temperature? Can you explain further ∆T?
Reference problems in figure 15 (should be 36 and 37) and ref 37 needs to me precise.
what is the different colours in figure 17. this figure, as many others, requires to be described and discussed.
Reorganize figure 19
Rewrite conclusion
As a conclusion, there are too many gaps into this manuscript to be considered for publication. The authors must improve the substance and the form.
The adjunction of PCM into cementitious materials impacts significantly the mechanical properties of the composite. The higher the amount of PCM, the higher the impact on the mechanical properties. This study makes sens if it is completed with some mechanical properties of the concrete at hardened state. Indeed, if the concrete does not meet the conditions of use once hardened, it is pointless to consider the use of PCM to avoir cracks and raptures when drying. This study should propose an optimum of PCM to use for that purpose.
The use of liquid PCM must also be precised (design and manufacturing of the composite). I believe that liquid PCM are not suitable as the may easily leak from the host material and they may have additional issues. The authors meus justify why they have chosen to use liquid PCM in their study.
Author Response
Thank you for your thorough review of the article. Our answers are in the file.

Round 2
Reviewer 3 Report
The introduction as well as the background still need to be improve in order to give the all picture of the context of the study as well as the state of the art about the topic.
The description of the results is still a little bit light. The reader has to do the effort of analysing the graph in general. Figures can be improved and overall, the figure captions are not very informative. It desserves a little bit more of information.
I am not an english native but I believe that the english still requires editing and style improvement. It has been improved but i do not think it was by an english native. If the sis the case, please clearly identify this person in the acknowledgments.
My comments are available directly on the pdf copy of the manuscript

Author Response
The authors thank to the reviewer for his very important help in improving of the article. All necessary information to reproduce both the experiments and simulations are now included in the article in our opinion. The additional explanations was also added in many points. Unfortunately, sometimes we had a problem with reading of the reviewer’s handwriting. However, we did our best to introduce all element to the article. The problem is also that we submit the current version of the article and after that Journal is using the comparison of the files function in MSOffice. If we modified the picture, two are appear in new file. Please look every time on the file submitted by authors (docx or pdf). The proofreading will be done if the editor wish to but in our opinion the text is clear.
The most important comments from the PDF are considered in file.
